# Physiologic Effects of Exogenous Dextrose in Murine *Klebsiella pneumoniae* Sepsis Vary by Route of Provision

**DOI:** 10.3390/nu12102901

**Published:** 2020-09-23

**Authors:** Byron Chuan, Lanping Guo, Bryce Cooper, Sagar Rawal, Teresa Gallego-Martin, Yingze Zhang, Bryan J. McVerry, Christopher P. O’Donnell, Faraaz Ali Shah

**Affiliations:** 1Division of Pulmonary, Allergy, Critical Care Medicine, University of Pittsburgh, Pittsburgh, PA 15261, USA; byc4@pitt.edu (B.C.); guol@upmc.edu (L.G.); bac89@pitt.edu (B.C.); sagarrawal@pitt.edu (S.R.); gallegomartint@upmc.edu (T.G.-M.); zhanyx@upmc.edu (Y.Z.); mcverrybj@upmc.edu (B.J.M.); odonnellcp@upmc.edu (C.P.O.); 2Acute Lung Injury Center of Excellence, University of Pittsburgh, Pittsburgh, PA 15261, USA; 3Center for Medicine and the Microbiome, University of Pittsburgh, Pittsburgh, PA 15261, USA; 4VA Pittsburgh Healthcare System, Pittsburgh, PA 15240, USA

**Keywords:** carbohydrate, inflammation, sepsis, pneumonia, hyperglycemia

## Abstract

Sepsis is characterized by a dysregulated immune response to infection. Nutrition is important in the care of septic patients, but the effects of specific nutrients on inflammation in sepsis are not well defined. Our prior work has shown benefits from early enteral dextrose infusion in a preclinical endotoxemia model of sepsis. In the current study, we extend our initial work to examine the effects of dextrose infusions, varying by route of administration, on inflammation and glycemic control in a more clinically relevant and translational model of *Klebsiella pneumoniae* (KP) bacteremia. Ten-week old C57BL6/J male mice (*n* = 31) underwent the implantation of indwelling vascular catheters, followed by inoculation with oropharyngeal KP. The mice were randomized 24 h after inoculation to (1) intravenous (IV) dextrose, (2) enteral dextrose, or (3) enteral saline (control) to study the effects on systemic inflammation, hemodynamics, and glycemic control. At 72 h, 77% of the control mice died, whereas IV dextrose induced 100% mortality, associated with increased inflammation, hyperglycemia, and hypotension. Enteral dextrose reduced mortality to 27%, promoted euglycemia, and reduced inflammation compared to IV dextrose. We conclude, in a bacteremic model of sepsis, that enteral (but not IV) dextrose administration is protective, suggesting that the route of nutrient support influences inflammation in sepsis.

## 1. Introduction

Sepsis induces a catabolic state characterized by metabolic dysregulation and muscle atrophy, contributing to an increased length of intensive care unit (ICU) stay, decreased quality of life, and increased long-term mortality [1,2,3,4]. Nutrition therapy can counteract the effects of catabolism in sepsis and improve recovery [5,6,7,8] but is potentially associated with adverse consequences.

Nutrition may be delivered to septic patients by either enteral or parenteral (intravenous (IV)) routes. While both are used in the setting of critical illness, studies have found route-associated risks in the use of either feeding method. Intravenous nutrition has been linked to an increased incidence of hyperglycemia (which is independently associated with increased mortality), increased risk of infection, and increased morbidity in septic patients [9,10,11]. Enteral nutrition also carries risk and has been associated with an increased incidence of diarrhea, nausea, vomiting, bowel ischemia, pulmonary aspiration, and hyperglycemia [12,13]. Full enteral nutrition, in particular, has been associated with high gastric residuals and an increased risk of aspiration [14,15]. Lower-level or trophic feeding has been shown to promote intestinal integrity, improve immune function, and prevent bacterial translocation with a lower risk of gastrointestinal side effects [16,17,18]. Recent studies have also suggested that overfeeding during critical illness may counter the beneficial effects of catabolism on enhancing autophagic processes that limit tissue injury and improve host resilience, thus favoring a trophic approach [19]. Current critical care guidelines recommend early trophic enteral nutrition over parenteral nutrition for most patients [20]. Interestingly, systematic reviews of trials published in the past two decades have shown that, whereas the use of enteral nutrition is associated with decreases in the incidence of infectious complications and length of ICU stay compared to IV nutrition, the route of nutrition support does not appear to affect overall mortality [5,21,22]. Better understanding of the effects of the route of nutritional support in critically ill patients has been hampered by the heterogeneity of study populations and variable quality in trial conduct [20,22], as well as a dearth of studies with a focus on the mechanisms of disease.

Carbohydrates are an important component in most nutrition formulations, but the effects of early carbohydrate provision on inflammation are not well characterized in the acute phase of sepsis. Our prior work characterized the effects of dextrose infusion varying by route of administration (enteral versus IV) on glycemic control and inflammation in a preclinical endotoxemia model. IV dextrose worsened glucose tolerance and insulin secretion and blunted insulin sensitivity, while enteral dextrose decreased proinflammatory cytokines and promoted euglycemia [23]. Endotoxemia provides an immediate and sterile model of inflammation, contrasting with the course of septic patients, who have an initial infectious insult that is often ongoing when systemic inflammation subsequently develops. Thus, the purpose of the current study was to understand the effects of dextrose provision in a translationally relevant model of sepsis. 

Pneumonia is the most common cause of sepsis but is not often utilized in murine septic models [24]. We developed a preclinical model of sepsis using oropharyngeal *Klebsiella pneumoniae* because (a) *Klebsiella* infection is common in septic patients, (b) the lung is a common route of entry for pathogens in clinical sepsis, (c) *Klebsiella pneumoniae* readily disseminates after initial pulmonary infection, and (d) systemic inflammation and decompensation develop approximately one day after the initial insult, thereby modeling the course of septic patients. We hypothesized that, in a *Klebsiella pneumoniae* model of sepsis, enteral dextrose provision would decrease inflammation, improve glycemic control, and lower mortality while IV dextrose at an equivalent dose would worsen inflammation, glycemic control, and increase mortality. 

## 2. Materials and Methods 

### 2.1. Animals

All experiments were performed in 10 to 12-week-old male C57BL/6J mice (Jackson Laboratory) in accordance with the Guide for the Care and Use of Laboratory Animals of the National Institutes of Health under protocols approved by the Institutional Animal Care and Use Committee (IACUC) at the University of Pittsburgh (Protocol Number: 16119554). The mice were housed individually during experimentation and given ad libitum access to autoclave-sterilized water and irradiated lab-grade regular chow (Labdiet Prolab RMH 3000 5P76; Purina, St. Louis, MO, USA) for the entire duration of the protocol. 

### 2.2. Experimental Model

All mice (*n* = 31) underwent arterial catherization simultaneous with either venous catheterization or gastric cannulation as previously described (Figure 1) [23]. Briefly, the mice were anesthetized under 2% inhaled isoflurane, and a 23-gauge Micro-Renathane catheter (MRE025; Braintree Scientific, Braintree, MA) was inserted into the right femoral artery, followed by the implantation of a 23-gauge Micro-Renathane catheter into the right femoral vein or an 18 gauge Micro-Renathane cannula (MRE050; Braintree Scientific, Braintree, MA, USA) into the gastric fundus. Catheters and cannulas were secured using sutures and glue, were tunneled subcutaneously to a small incision between the scapulae where they exited the body, and were connected to a dual-channel swivel (375/D/22QM; Instech, Plymouth Meeting, PA, USA) that allowed free movement upon recovery. Immediately following catheterization and while still under the effects of anesthesia, mice were inoculated with oropharyngeal *Klebsiella pneumonia* (KP) strain ATCC 43816 at 2 × 10^4^ colony forming units (CFUs) in 100 µL of saline. 

Heparinized saline (20 units/mL) was continuously infused at 7 µL/h through catheters and gastric cannulae to maintain patency. Blood pressure (BP) was measured continuously using the WinDaq Acquisition Software (v2.94; DATAQ Instruments, Akron, OH, USA). Twenty-four hours after inoculation, blood was sampled with an arterial catheter (~50 µL), followed by randomization to receive (1) enteral saline at 100 µL/h in the control group, (2) IV dextrose (50% dextrose (D50)), or (3) enteral dextrose (D50). This rate of D50 infusion (which provides ~40% of daily caloric needs) had previously induced consistent and reproducible effects on systemic inflammation and glucose metabolism in lipopolysaccharide (LPS) and cecal ligation and puncture models [23,25,26]. Additionally, 50% dextrose is a readily available concentration that provides the caloric targets in our model while minimizing the volume of infusion. Our study focused on KP-exposed groups, informed by our prior studies that did not reveal significant differences in vehicle-exposed or IV saline-exposed groups. [23]. Blood was sampled with an arterial catheter again at 48 and at 72 h post-inoculation (hpi). All blood samples were tested for blood glucose before centrifugation for plasma collection. If an animal’s mean arterial BP reading dropped below 60 mmHg (corrected value, described in greater detail below) for more than 1 h prior to the end of the experiment, the mouse was anesthetized and underwent terminal sacrifice. We chose this threshold because in a pilot set of studies, blood pressure below 60 mmHg for at least one hour was predictive of impending death in 100% of cases (unpublished data). Mice were sacrificed at 72 hpi.

### 2.3. Bronchoalveolar Lavage (BAL)

Prior to sacrifice, the mice were anesthetized with 2% isoflurane, followed by insertion of a 20-gauge IV catheter (3066; Smiths Medical, Minneapolis, MN, USA) into the trachea. A length of nylon suture was used to tie off the left bronchus, and 2 mL of phosphate-buffered saline (PBS) + 0.6 mM EDTA was lavaged in four 500 µL aliquots into the right lung (the total volume of return was 1.7 mL). 10% buffered formalin was then administered into the right lung after BAL and held at a pressure of 15 cm H_2_O, and fixed lungs were subsequently embedded in paraffin blocks for sectioning. The BAL was spun at 2000 rpm for 5 min at 4 °C. The supernatant was separated and aliquoted for subsequent analyses, and the cell pellet was resuspended for cell counting using an automatic cell counter (TC-10; Bio-Rad Laboratories, Hercules, CA, USA). 

### 2.4. Assessment of Bacterial Burden

A portion of the left lung and whole spleen were weighed, homogenized in 1 mL of PBS, serially diluted, plated on tryptic soy agar plates, and incubated at 37 °C. Bacterial CFU counts were performed after 24 h, and the results were normalized to tissue mass (in mg).

### 2.5. Quantification of Hemodynamic Data

Files containing raw Windaq data were analyzed using LabChart Reader (ver.8; ADInstruments, Colorado Springs, CO, USA). A correction of +41 mmHg was applied to each BP waveform to account for a 21.94 inch difference between the mice and the pressure transducer. The mean arterial pressure (MAP) was calculated as 1/3 systolic BP + 2/3 diastolic BP. The heart rate was calculated from waveforms, counting the number of peaks in a BP waveform using a manually specified threshold. Hemodynamic data are presented and were analyzed using hourly averages and were censored when the BP fell below the 60 mmHg threshold for 1 h; thus, the hemodynamic data shown represent the surviving mice at each timepoint, not all mice in the experiment. 

### 2.6. Biochemical Assays 

Arterial blood glucose was measured every 24 h post-inoculation using a handheld glucometer (Solus V2; Biosense Medical, Chelmsford, UK). Glucose measurements above the limit of detection were assigned a value of 600 mg/dL (the upper limit of the glucometer). Plasma insulin was measured via ELISA (Ultra-Sensitive Mouse Insulin ELISA Kit 90080; Crystal Chem Inc., Elk Grove Village, IL, USA). BAL protein concentrations were measured in supernatants using a Pierce BCA Protein Assay Kit (23225; Thermo Fisher Scientific, Waltham, MA, USA). Plasma cytokines (TNF-α, IL-1β, IL-6, IL-10, G-CSF, and MCP-1) were measured using a multiplex assay (M60-009RDPD, Bio-Rad Laboratories, Hercules, CA, USA). 

### 2.7. Lung Injury Scoring

A longitudinally cut 10 µm-thick section from the right lung was stained with hematoxylin and eosin. Ten random fields from each slide were imaged on a bright field microscope (Provis AX700; Olympus, Shinjuku, Tokyo, Japan) using a 60× objective. Lung injury was quantified by two independent and blinded researchers using a modified Murray Scoring system, where a score of 1–4 was assigned based on the amount of cellular infiltration, obstruction of airways, and visible space in the lung parenchyma, with higher scores corresponding with worse lung injury [27]. An overall lung injury score for each mouse was assigned using the average for each of the 10 images.

### 2.8. Statistical Analysis 

Differences in survival were evaluated using a log-rank test. Hemodynamic data were analyzed using two-way ANOVA with post-hoc Tukey’s multiple-comparison tests. Data on bacterial burden, lung injury, and inflammation were analyzed using one-way ANOVA with post-hoc Tukey’s multiple-comparison tests. Contingency tables were used to analyze associations between mortality at 72 hpi and glycemic state; significance was evaluated using Fisher’s exact test. Differences were considered statistically significant at *p* < 0.05.

## 3. Results

### 3.1. Physiologic Effects of Early Dextrose Infusion Vary by Route of Administration in a KP Model of Sepsis 

Mice receiving enteral saline in our model had a survival rate of 23% at 72 hpi (Figure 2). IV dextrose infusion, started at 24 hpi, decreased survival with no mice alive at 72 hpi. In contrast, enteral dextrose at an equivalent level increased survival to 73% at 72 hpi (*p* < 0.001).

In the first 24 hpi, prior to the initiation of saline or dextrose administration, MAP was similar between the three groups (Figure 3A,B). Mice receiving enteral saline experienced a gradual decline in MAP starting at 24 hpi, eventually becoming hypotensive by 72 hpi. IV dextrose infusion accelerated hemodynamic collapse, with significantly more hypotension from 24 to 48 hpi compared to that with enteral saline infusion (*p* < 0.001). MAP was similar between mice receiving enteral saline and enteral dextrose during the 24–48 hpi period. From 48 to 72 hpi, MAP was higher in mice receiving enteral dextrose compared mice receiving enteral saline (*p* = 0.016) despite both groups exhibiting a decrease in MAP from 24 to 48 hpi. 

Patterns of heart rate over time were similar to those of the MAP, despite mice in the IV dextrose group unexpectedly having a lower initial heart rate (HR) compared to the enteral saline and enteral dextrose groups (Figure 3C,D). Mice receiving enteral saline maintained a normal HR from 24 to 48 hpi but displayed significantly reduced HR by 72 hpi. Consistent with the pattern for MAP, IV dextrose infusion induced an early drop in HR from 24 to 48 hpi, whereas mice receiving enteral dextrose maintained a normal HR for the entire duration of the experiment (enteral dextrose vs. enteral saline on Day 3, *p* < 0.001). 

### 3.2. IV Dextrose Worsens Inflammation and Lung Permeability in a KP Model of Sepsis

Systemic cytokines were similar between groups at 24 hpi, prior to the initiation of experimental treatments. Infusion of IV dextrose increased inflammation at 48 hpi, with elevations in IL-6, IL-10, MCP-1, and G-CSF compared to mice receiving enteral saline or enteral dextrose (Figure 4). Cytokine levels were similar between enteral dextrose and saline-infused groups throughout the duration of the experiment. Differences were not noted for the cytokines IL-1β and TNF-α at the timepoints measured in this study (Figure 4A,B). IV dextrose infusion increased BAL total protein levels compared to those in enteral saline and enteral dextrose groups (Figure 5A; *p* = 0.031 and *p* = 0.003, respectively), accompanied by trends for increases in the BAL cell count and BAL neutrophil proportion (Figure 5C,D; *p* = 0.093 and *p* = 0.108, respectively). No significant differences were detected in BAL total protein, BAL cell count, BAL neutrophils, or histological lung injury scores between the enteral dextrose and saline groups (Figure 5). 

### 3.3. Exogenous Dextrose Does Not Affect Bacterial Burden in a KP Model of Sepsis

Infusion of dextrose did not increase bacterial burden, whether administered enterally or parenterally. No differences in lung CFUs were noted between any of the three groups at the time of sacrifice, indicating similar bacterial burdens (Figure 5E). Similarly, no differences were observed in the splenic CFUs between the three groups, consistent with a comparable degree of bacterial dissemination under each of the three experimental conditions (Figure 5F).

### 3.4. Improvement in Survival with Enteral Dextrose Is Associated with Preservation of Euglycemia

KP-exposed mice receiving enteral saline developed hypoglycemia between 24 and 72 hpi, with no increase in circulating insulin (Figure 6). IV dextrose infusion initiated at 24 hpi induced severe hyperglycemia by 48 hpi with a concurrent increase in plasma insulin; the heightened circulating insulin levels paired with unresolved hyperglycemia is consistent with peripheral insulin resistance. By contrast, infusion of enteral dextrose promoted euglycemia at 48 hpi and 72 hpi and was associated with an increase in insulin comparable to that observed with IV dextrose (*p* = 0.140).

Preservation of euglycemia was associated with survival (Table 1). Of the nine mice receiving enteral saline, seven progressed to hypoglycemia and died before 72 hpi and two remained euglycemic and survived to 72 hpi (*p* = 0.028). Of the combined 22 mice in the two dextrose infusion groups, 15 (all 11 mice receiving IV dextrose and four mice receiving enteral dextrose) progressed to hyperglycemia, with 14 dying by 72 hpi, while seven maintained euglycemia and survived to 72 hpi (*p* < 0.001).

## 4. Discussion

In a translationally relevant *Klebsiella pneumoniae* model of sepsis, the physiologic effects of exogenous dextrose varied by route of administration. Intravenous dextrose decreased survival and worsened systemic inflammation as evidenced by increases in IL-6, IL-10, MCP-1, and G-CSF, concurrent with markedly decreased MAP and HR. IV dextrose increased lung permeability, though it did not increase the bacterial burden in the lung or spleen. Lastly, IV dextrose induced severe glucose dysregulation, manifesting as hyperglycemia associated with an increase in insulin that was unable to control blood glucose, inferring a state of insulin resistance. Thus, increased mortality in KP-exposed mice treated with IV dextrose was associated with a combination of increased systemic inflammation and adverse effects of hyperglycemia. 

The activation of pro-inflammatory pathways can negatively impact insulin sensitivity and may drive the hyperglycemia observed with the IV dextrose intervention [28,29,30]. Hyperglycemia, in turn, has been shown to exacerbate the pathogenic effects of infectious agents through further promotion of the inflammatory response, decreased neutrophil efficiency, and the inhibition of effective phagocytosis and opsonization [31,32,33]. The relationship between inflammation, insulin resistance, and hyperglycemia has been well-documented in human studies, where the excessive inflammatory response under septic conditions creates a state of insulin resistance, leading to hyperglycemia, which is independently associated with increased mortality [9,10,11,34]. Systemic inflammation and hyperglycemia likely amplify each other in the setting of IV dextrose administration, resulting in the significantly higher mortality seen in the IV dextrose group. Hypotension can also be seen as a marker of inflammatory activity levels in sepsis, wherein sufficiently high inflammatory activity causes vasodilation and the systemic lowering of blood pressure, progressing to septic shock [35]. Indeed, both MAP and HR decreased sharply after the start of IV dextrose infusion. Taken together, our findings suggest the hemodynamic collapse and increased mortality in KP-exposed mice are secondary to the exaggerated systemic inflammation and hyperglycemia induced by IV dextrose.

The findings in the KP model differ from our prior studies using an LPS) sterile model of sepsis. After a 5 h exposure to LPS, mice receiving IV dextrose did not demonstrate increases in proinflammatory cytokines compared to saline-infused mice, and while decreases in insulin sensitivity were observed, the mice did not develop overt hyperglycemia. In the KP model, the longer time course of a sustained bacterial insult allows for the establishment of persistent hyperglycemia; as hyperglycemia has been shown to exacerbate inflammation [36,37], it is possible that our LPS model was too acute for these effects to manifest. Notably, since mice receiving IV dextrose were sacrificed at ~48 hpi for ethical considerations due to the severity of illness, the interpretation of findings from the terminal samples (such as BAL) compared to those from the other groups is somewhat limited due to the differences in timing post-infection. However, within the context of these limitations, our results suggest that the administration of IV dextrose is harmful in the setting of KP sepsis. 

By contrast, enteral dextrose improved survival in a KP model without reductions in systemic inflammation. KP-exposed mice receiving enteral dextrose showed proinflammatory cytokine levels similar to those of control mice receiving saline, along with equivalent bacterial burden, lung injury, and lung permeability. The infusion of enteral dextrose promoted euglycemia, which may be the primary driving factor behind the observed survival benefit. Euglycemia in mice receiving enteral dextrose was associated with the prevention of hemodynamic collapse, as opposed to the bradycardia and hypotension seen in the other groups. Our hemodynamic data do have the potential for survivor bias; however, we tried to limit this effect by censoring data for sicker mice once the BP dropped consistently below a threshold that was predictive of death so as not to exaggerate differences between groups. Examination of the association between glucose control and mortality across all mice demonstrated that both hypoglycemia and hyperglycemia were significantly associated with mortality, findings that are consistent with several preclinical and clinical studies [38,39]. Our prior studies in an LPS model suggested that beneficial effects of enteral dextrose in sepsis may be exerted through an anti-inflammatory effect, but we did not observe decreases in cytokine levels with enteral dextrose in a KP model. Notably, cytokines were only measured every 24 h in this study, and it remains possible that transient changes in the inflammatory response were not captured by the sampling protocol. 

Since enteral dextrose administration in a KP model was associated with improved survival without decreases in inflammation or in bacterial burden, our findings suggest that enteral dextrose potentially exerts its beneficial effects through its promotion of disease tolerance, a concept that refers to the host’s ability to maintain physiologic homeostasis during an infectious insult [40]. By minimizing the damage done to tissues that promote homeostasis, the pathogenic effects of the infectious agent are reduced, despite no direct effect on the pathogen load itself. For example, in a recent study conducted by Weis et al. [41], the investigators examined the role of ferritin H chain (FTH) in establishing disease tolerance and demonstrated several key points relevant to our findings. In both their cecal ligation and puncture (CLP) and this KP model, hypoglycemia was significantly associated with increased mortality, while the maintenance of blood glucose in a euglycemic range promoted survival. Additionally, despite differences in survivorship in both models, differences in systemic inflammation, bacterial load, or tissue injury were not observed between groups. In their model, the administration of exogenous FTH promoted liver gluconeogenesis to combat the hypoglycemia caused by the anorexia of the infection, thus maintaining blood glucose in a euglycemic range and promoting host resilience against the infection, similar to the effects of enteral dextrose in our model. 

The current clinical paradigm regarding the route of nutrition support in sepsis is that enteral feeding is the preferred route, though it is associated with relative nutritional inadequacy due to feed intolerance and gastric complications. IV nutrition, while thought to deliver nutrition more efficiently, is associated with an increased risk of hyperglycemia and infectious complications [42,43,44,45,46]. In recent years, this paradigm has been challenged, with meta-analyses demonstrating a lack of differences in patient outcomes between the two feeding strategies [5]. Results from clinical studies are somewhat challenging to interpret, secondary to the heterogeneity of the study populations and the lack of standardization of nutritional support volume, timing, and formulations. A solid preclinical base establishing the mechanistic effects of nutrients and the route of nutrition in the acute phase of sepsis could inform future clinical and translational studies. Our preclinical data reinforce that the route of delivery of nutrition may be a major determinant in the progression of sepsis; IV dextrose precipitates an excessive inflammatory response and the dysregulation of glucose metabolism, leading to hypotension, bradycardia, and increased mortality in a translationally relevant KP model. Enteral dextrose administration confers protective effects in the setting of sepsis, specifically in glucose regulation and hemodynamic homeostasis, promoting overall survival through mechanisms that promote host resilience rather than reducing systemic inflammation. Further preclinical and clinical studies are needed to uncover the mechanisms underlying the promotion of host resilience through the provision of enteral support in sepsis.

## Figures and Tables

**Figure 1 nutrients-12-02901-f001:**
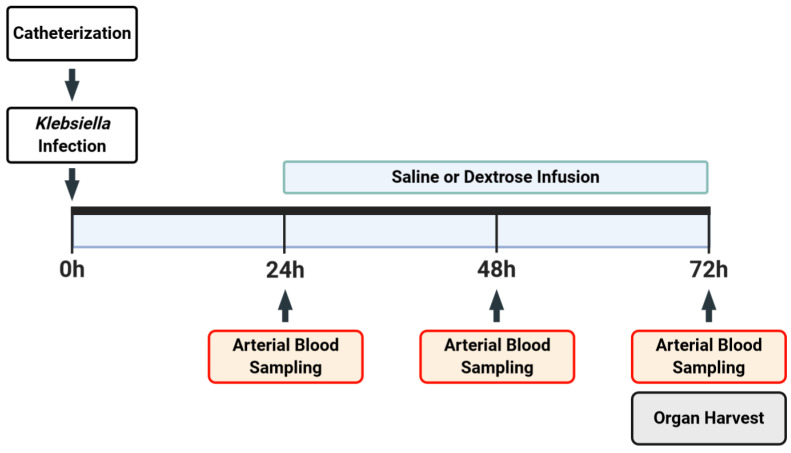
Experimental protocol. All mice (*n* = 31) received femoral arterial catheterization accompanied by either femoral venous catherization or gastric cannulation, followed immediately by inoculation with oropharyngeal *Klebsiella pneumoniae*. Study infusion was started 24 hpi. Arterial blood was sampled every 24 h for assessment of glucose, insulin, and cytokine levels.

**Figure 2 nutrients-12-02901-f002:**
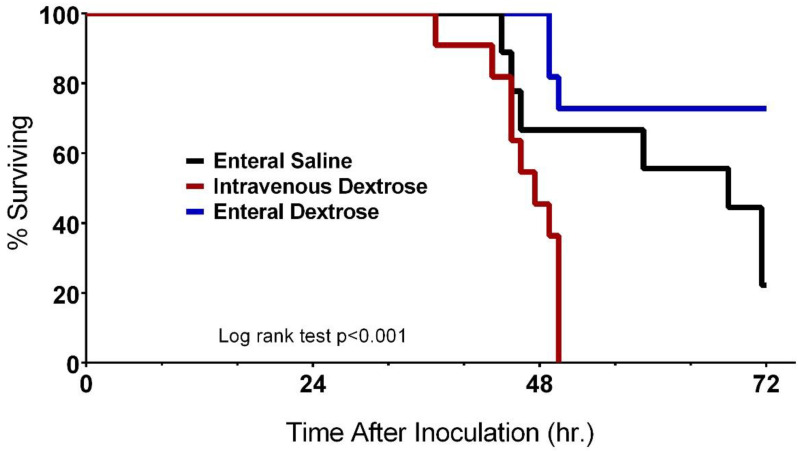
Effects of dextrose provision on survival following *Klebsiella pneumoniae* infection vary by route of provision. Mice receiving intravenous (IV) dextrose (*n* = 11) had decreased survival compared to mice receiving enteral saline (*n* = 9), but survival was increased in mice receiving enteral dextrose (*n* = 11, *p* < 0.001).

**Figure 3 nutrients-12-02901-f003:**
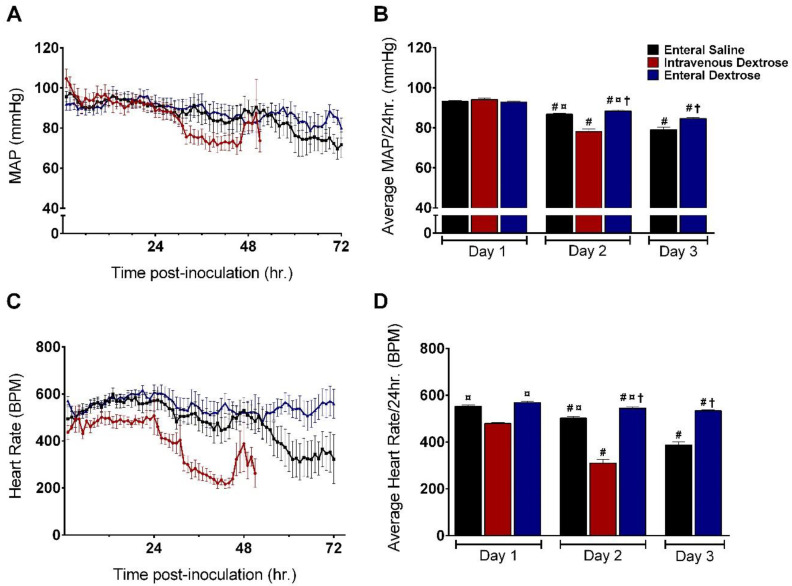
Hemodynamic effects of exogenous dextrose in *Klebsiella pneumoniae* (KP) infection. (**A**) Average mean arterial pressure (MAP) by hour following exposure to KP. Values represent data for surviving mice; mice are censored at the time of death or at the time that MAP fell below the specified threshold (60 mmHg) for one hour. (**B**) Average daily MAP by infusion group. # = *p* < 0.05 compared to average MAP on prior day within same group; ⸋ = *p < 0.05* compared to IV dextrose on same day; † = *p* < 0.05 compared to enteral saline group on same day. (C) Average heart rate (HR) by hour following KP exposure. (D) Average daily HR by infusion group. # = *p* < 0.05 compared to average MAP on prior day within same group; ⸋ = *p <* 0.05 compared to IV dextrose on same day; † = *p* < 0.05 compared to enteral saline group on same day. Abbreviations: BPM: beats per minute.

**Figure 4 nutrients-12-02901-f004:**
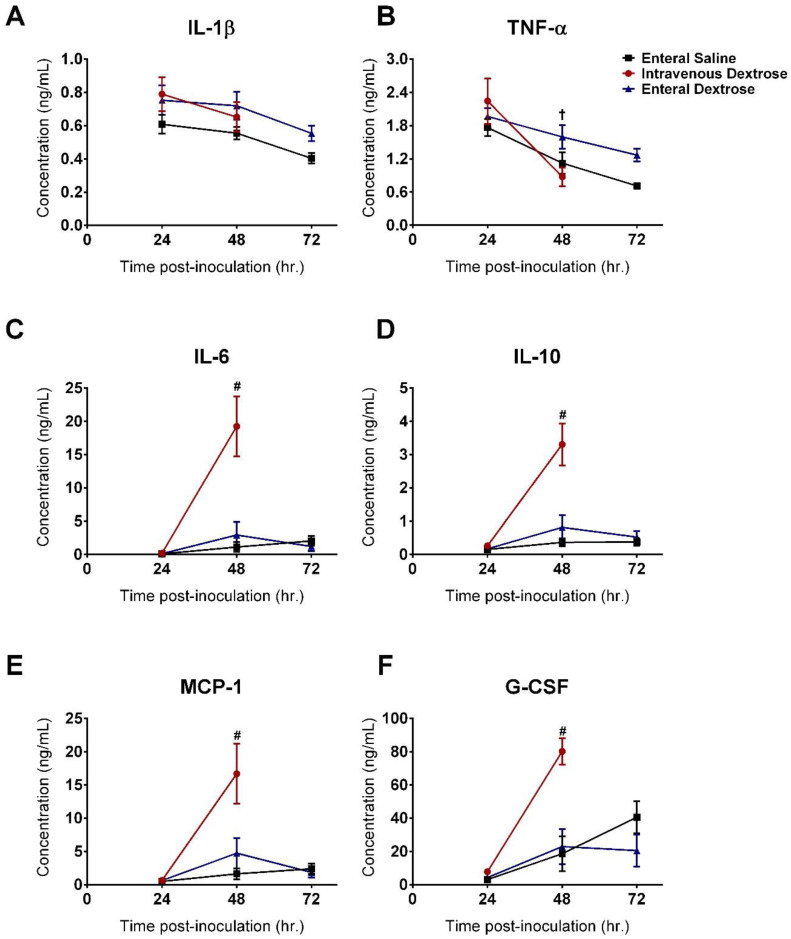
IV but not enteral dextrose worsened systemic inflammation in a murine KP model. (**A**) IL-1β. (**B**) TNF-α. (**C**) Il-6. (**D**) IL-10. (**E**) MCP-1. (**F**) G-CSF. Two-way ANOVA with Tukey’s multiple-comparison tests was used to compare groups at all timepoints. # = IV dextrose compared to enteral saline or enteral dextrose at 48 hpi, *p* < 0.01; † = IV dextrose levels compared at 24 and 48 hpi, *p* = 0.002. Abbreviations: IL-1β: Interleukin 1 beta. TNF-α: Tumor necrosis factor alpha. IL-6: Interleukin 6. IL-10: Interleukin 10. MCP-1: Monocyte Chemoattractant Protein-1. G-CSF: Granulocyte colony-stimulating factor.

**Figure 5 nutrients-12-02901-f005:**
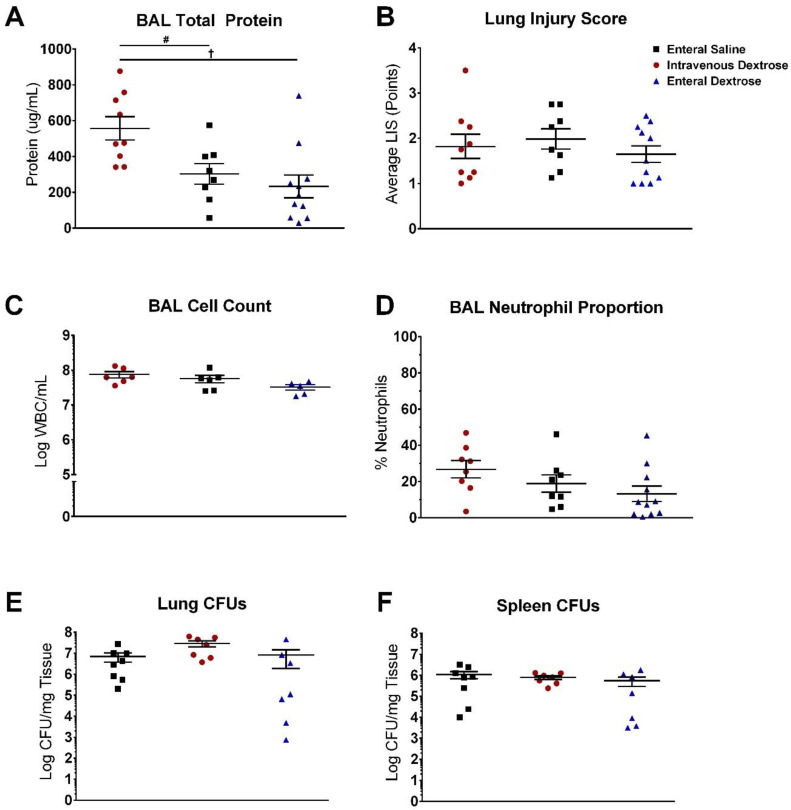
IV dextrose increases lung permeability in a KP model without increases in bacterial burden. (**A**) Bronchoalveolar lavage (BAL) protein concentration measured from BAL supernatant. (**B**) Histological assessment of hematoxylin and eosin-stained sections of the right lung. (**C**) Total BAL cell counts. (**D**) Proportion of neutrophils in BAL supernatant. (**E**) Colony forming units (CFUs) from whole left lung homogenate. (**F**) CFUs from splenic homogenate. # = *p* < 0.05 for comparison between IV dextrose and enteral saline groups. † = *p* < 0.05 for comparison between IV dextrose and enteral dextrose groups by one-way ANOVA with Tukey’s multiple-comparison tests. Abbreviations: BAL, bronchoalveolar lavage. LIS: lung injury score. WBC: white blood cell.

**Figure 6 nutrients-12-02901-f006:**
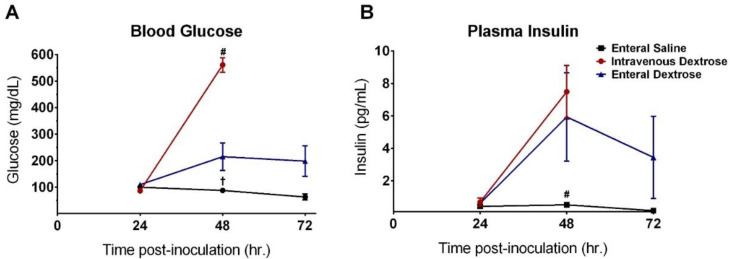
Enteral dextrose infusion promotes euglycemia in septic mice. (**A**) Blood glucose measured at 24 h intervals post-inoculation. (**B**) Plasma insulin measured at 24 h intervals post-inoculation. Two-way ANOVA was used to compare groups at all timepoints with Tukey’s multiple-comparison test. # = IV dextrose compared to enteral dextrose at 48 hpi, *p* < 0.01. † = Enteral saline compared to enteral dextrose at 48 hpi, *p* = 0.015.

**Table 1 nutrients-12-02901-t001:** Euglycemia is associated with survival in a murine KP model.

	Saline Infused			Dextrose Infused	
	Euglycemic	Not Euglycemic		Euglycemic	Not Euglycemic
Survived	2	0	Survived	7	1
Died	0	7	Died	0	14
	*p*-value	0.028		*p*-value	<0.001

Mice in saline or dextrose-infused groups were categorized as euglycemic or not euglycemic based on the level of glycemic control during the 72 h experiment. Associations between survival and glycemic status were assessed using Fisher’s exact test.

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
