# Peer review of "Physiologic Effects of Exogenous Dextrose in Murine Klebsiella pneumoniae Sepsis Vary by Route of Provision"

_nutrients, 2020, doi:10.3390/nu12102901_

Round 1

Reviewer 1 Report

In this study, Chuan, B. et al. investigated the effects of IV dextrose and enteral dextrose in a bacteria-induced sepsis. The research group has published the effects of IV dextrose and enteral dextrose in LPS-induced sepsis. In this study, they used Klebsiella pneumoniae model. In this study, they concluded that enteral dextrose is beneficial for maintenance of glycose and preservation of euglycomia was associated with survival. The finding in this preclinical study provides important evidence to show enteral dextrose administration is beneficial for protecting against sepsis.

Concern #1: The authors need to explain why 50% of dextrose was selected.

Concern #2:  The data in the table I is important for the conclusion in this study. The authors should show the statistical data in the Table 1.

Concern #3: I am confused about the BAL samples in the Fig. 5. Are the BAL samples of IV dextrose group from the 48hpi? It it is the case, is there any rationale for comparing these samples to other groups from 72hpi?

Reviewer 2 Report

In this work the authors attempt to explore the physiologic effects and outcome differences caused by enteral vs parenteral dextrose nutrition in a murine sepsis pneumonia model. Strengths of the present work include using a more robust and translational sepsis model than previous studies. Additionally, this works seems to have informed, at least in part, the basis of a human pilot study (NCT03454087; DOI: 10.1002/jpen.1608) that recently completed but for which results are not yet available. Overall, this work adds to the body of literature regarding benefits of early enteral nutrition in sepsis and may be viewed with interest. However, the present work does little to address and discuss its potential limitations and biases. Comments are provided to help improve this discussion.

  • Suggest adding discussion of the proposed benefits of catabolism and underfeeding in sepsis to the introduction (eg, https://doi.org/10.1186/s13054-020-2771-4)
  • In the author’s previously cited model (reference 17 of the present work) there was a 3-day recovery period allotted between the implantation of the IV and gastric catheters and the initiation of the sepsis model. What was the rationale for not including that recovery period in the present model?
  • Suggest discussing the “peaks” in BP and HR seen in the IV dextrose group around 48h of figure 3 in the discussion.
    • The authors mention on line 130-131 that “hemodynamic data shown is represents the surviving mice at each timepoint, not all mice in the experiment.” Was this this just an artifact of survivor bias, and that as the more sick mice died and were censored, the mean increased briefly before falling again?
  • There may be a time bias in the results reported in figure 5. That is, as the IV dextrose group died near 48hpi and most of the enteral dextrose group survived until sacrifice at 72hpi. The infection characteristics shown in figure 5 may have improved or worsened in the enteral nutrition group over the 48-72hpi time. This bias is easily understood in figure 4, for example, as there are no measurements for the 72hpi timepoint. This disparity in data in Figure 5 was caused by ethical animal care and allowing for greater duration of data generation in survivors, but the imbalanced assessment time must be discussed as a limitation of the study as it does not necessarily represent direct effects of treatment in these single timepoint samples.
